# Efficacy of Combination Chemotherapy Using a Novel Oral Chemotherapeutic Agent, FTD/TPI, with Ramucirumab Murine Version DC101 in a Mouse Syngeneic Cancer Transplantation Model

**DOI:** 10.3390/jcm9124050

**Published:** 2020-12-15

**Authors:** Kenta Tsunekuni, Hisato Kawakami, Kazuaki Matsuoka, Hideki Nagase, Seiichiro Mitani, Kazuhiko Nakagawa

**Affiliations:** 1Discovery and Preclinical Research Division, Taiho Pharmaceutical Co. Ltd., Tsukuba, Ibaraki 300-2611, Japan; kenta-tsunekuni@taiho.co.jp (K.T.); kazu-matsuoka@taiho.co.jp (K.M.); 2Department of Medical Oncology, Faculty of Medicine, Kindai University, 377-2 Ohno-higashi, Osaka-Sayama, Osaka 589-8511, Japan; 192741@med.kindai.ac.jp (S.M.); nakagawa@med.kindai.ac.jp (K.N.); 3Discovery and Preclinical Research Division, Taiho Pharmaceutical Co. Ltd., Tokushima, Tokushima 771-0194, Japan; h-nagase@taiho.co.jp

**Keywords:** trifluridine/tipiracil, colorectal cancer, combination drug therapy, ramucirumab, mouse syngeneic transplantation

## Abstract

Trifluridine/tipiracil (FTD/TPI) (a.k.a. TAS-102) is a combination drug for metastatic colorectal cancer (CRC) and severely pretreated metastatic gastric/gastroesophageal junction (GEJ) cancers, comprising FTD, a thymidine-based antineoplastic nucleoside analog, and TPI, which enhances FTD bioavailability. Herein, in *KRAS* mutant murine colorectal cancer CT26 syngeneic models, we investigate whether combination therapy with DC101 (a surrogate ramucirumab antibody, rat antimouse vascular endothelial growth factor receptor (VEGFR)-2 monoclonal antibody (mAb)) improves FTD/TPI efficacy. Tumor growth inhibition (TGI) on day 15 was 38.0% and 30.6% upon DC101 monotherapy and FTD/TPI monotherapy respectively, and 60.3% upon combination therapy. Tumor volume was significantly lower (*p* < 0.001) upon combination treatment than upon FTD/TPI or DC101 monotherapy, indicating the additive effects of FTD/TPI and DC101. DNA-incorporated FTD levels on Day 8 were significantly higher in combination therapy with FTD/TPI (for 5 consecutive days) and DC101 (on alternate days for 7days) than in FTD/TPI monotherapy. Furthermore, vascular endothelial cell-specific marker CD31 was downregulated in DC101-treated tumors on day 8. These results indicate that combination therapy with FTD/TPI and DC101 is a promising treatment alternative regardless of *KRAS* mutations.

## 1. Introduction

Trifluridine/tipiracil (FTD/TPI), also known as TAS-102, is a combinational drug used for patients with metastatic colorectal cancer (CRC) [1]. The structure of FTD/TPI is depicted in Figure 1. It is a mixture of two drugs, FTD and TPI, at a molar ratio of 1:0.5. FTD is an analog of thymidine, which exhibits antitumor effects through two different mechanisms, inhibiting thymidylate synthase [2] and incorporating into DNA, thereby inducing DNA dysfunction [3]. TPI enhances the bioavailability of FTD by inhibiting its enzymatic degradation in the liver by inhibition of thymidine phosphorylase. TPI is therefore beneficial in producing a more durable and sustained response to FTD [4]. The main antitumor mechanism of FTD/TPI in the preclinical colon cancer model or in CRC is thought to be the incorporation of FTD into DNA [5]. FTD/TPI significantly improved overall survival and the safety profile in patients with metastatic CRC that is refractory to standard chemotherapies in the phase III study (RECOURSE) [6]. For gastric cancer, FTD/TPI has been used to treat patients with heavily pretreated advanced gastric/gastroesophageal junction (GEJ) cancers who have progressed or are intolerant to previous lines of therapy; it was found to improve overall survival (TAGS) [7].

Currently, various combination therapies have been attempted to enhance the therapeutic efficacy of FTD/TPI, particularly combination treatment with bevacizumab, which is considered a promising approach. Tumors have a poorly organized and hyperpermeable vasculature and hence have a diminished blood supply [8]. Bevacizumab antagonizes vascular endothelial growth factor (VEGF) and may normalize tumor vessels, thus improving tumor blood supply and increasing FTD accumulation in the tumor. A preclinical study reported that combination therapy with FTD/TPI and bevacizumab was more effective than either monotherapy in colorectal xenograft models developed from human cell lines SW48 (harboring wild-type *KRAS*) and HCT 116 (harboring mutant *KRAS*), and FTD levels in tumors were higher upon combination treatment than with FTD/TPI monotherapy [9]. Furthermore, combination treatment with FTD/TPI and bevacizumab among metastatic CRC patients refractory to standard therapies exerted promising effects with manageable safety in a phase I/II clinical trial, regardless of *KRAS* status [10] and in fact, bevacizumab showed an additive effect to FTD/TPI monotherapy in a Denmark study [11]. Based on these findings, combination treatment with FTD/TPI and other antiangiogenic agents such as ramucirumab, an anti-VEGF receptor (VEGFR)-2 antibody, also seems promising in advanced gastric/GEJ cancers. This study aimed to investigate the efficacy of combination therapy with FTD/TPI and antiangiogenic agents among gastric cancer patients; however, bevacizumab has not been approved for gastric cancer treatment. Therefore, tumor angiogenesis targeting ramucirumab, which has been approved for treating gastric cancer, is a promising drug among gastric cancer patients in combination with FTD/TPI. Herein, we assessed the efficacy of combination therapy with FTD/TPI and DC101, an antimouse VEGFR-2 monoclonal antibody, which inhibits VEGFR-2 activity like ramucirumab and is used as a surrogate antibody for ramucirumab according to the Eli Lilly’s interview form, and we evaluated the association between this additive effect and the levels of FTD uptake into DNA and markers associated with tumor immunity.

## 2. Experimental Section

### 2.1. Cell Lines and Culture Conditions

The mouse colon cancer cell line CT26 was obtained from American Type Culture Collection and grown in RPMI-1640 Medium (Sigma-Aldrich, Tokyo, Japan) supplemented with 10% fetal bovine serum (Hyclone, Waltham, MA, USA), 100 U/mL penicillin, and 100 µg /mL streptomycin (Life Technologies). Cells were grown at 37 °C in a humidified atmosphere with 5% CO_2_. All experimental procedures were performed using cells in the exponential growth phase.

### 2.2. Chemicals and Antibodies

FTD and TPI were obtained from Taiho Pharmaceutical Co., Ltd. (Tokyo, Japan). DC101 was purchased from BioXcell (Lebanon, NH, USA).

### 2.3. Animals

Male BALB/cAJcl mice were purchased from CLEA Japan (Tokyo, Japan) and housed under specific pathogen-free conditions, with food and water provided ad libitum. All animal studies were performed in accordance with the instructions of and with the approval of the Institutional Animal Care and use Committee of Taiho Pharmaceutical Co., Ltd.

### 2.4. Antitumor Activity In Vivo

CT26 tumor cells (approximately 5 × 10^6^ cells/mouse) were transplanted subcutaneously into the dorsal region of each mouse. Nine days later, the animals were grouped and randomized by tumor volume (determined using the following equation: 0.5 × length × width × breadth) uniformly in all groups on day 0. Each group consisted of six mice. FTD/TPI was prepared by mixing FTD and TPI at a molar ratio of 1:0.5 in 0.5% hydroxypropyl methylcellulose (HPMC) solution. The dose of FTD/TPI was expressed based on FTD content. FTD/TPI was administered orally from days 1 to 5 and 8 to 12 at the reported effective dose (200 mg/kg-bwt/day) [12]. DC101 was administered intraperitoneally (ip) at the reported effective dose of 0.8 mg into each mouse on days 1, 3, 5, 7, 9, 11, and 13 [13]. Treatment groups contained a similar range of initial tumor volumes (50 to 200 mm^3^ at day 0). For the control group, 10 mL/kg of vehicle (0.5% HPMC solution) was administered orally from days 1 to 5 and 8 to 12; and 0.1 mL of saline was injected intraperitoneally on days 1, 3, 5, 7, 9, 11, and 13.

To evaluate tolerability, body weight change (BWC) was determined using the following formula: BWC (%) = [(body weight on measured day) − (body weight on day 0)]/(body weight on day 0) × 100. Intolerability was defined as a BWC indicating weight loss of >20% or toxic death. The experimental endpoint was defined as the day on which the individual tumor volume in the individual body weight within each group approached >10%.

### 2.5. Flow Cytometry Analysis of Tumor-Infiltrating T Cells

Resected tumor tissues were cut into small pieces and digested at 37 °C for 20 min in RPMI-1640 supplemented with 0.05% collagenase Type IV (Worthington Bio-chemical Corporation, NJ, USA), 0.002% DNase I (Sigma-Aldrich Japan, Tokyo, Cat 10104159001 grade II), and 10% fetal bovine serum (Sigma-Aldrich, Japan). Dissociated cells were then filtered through a 100-μm mesh, using 10 mL of RPMI-1640 supplemented with 10% fetal bovine serum (Sigma-Aldrich) and washed twice with Ca/K-free phosphate-buffered saline (PBS) for 10 min at 400× *g* at room temperature. To determine the percentage of CD4, CD8 T cells, and tumor-infiltrating tumor-associated macrophages (TAMs), we suspended the residual cell pellet in MACS buffer and added mouse CD45 microbeads (Miltenyi Biotec, Bergisch-Gladbach, Germany). Thereafter, we rinsed the suspension using an MS column (Miltenyi Biotec) in accordance with the manufacturer’s instructions. The harvested cells were placed on a 96-well U-bottom plate, and each antibody (biolegend) was added (for determining CD4, CD8 T cells, PerCP-Cy5.5 antimouse CD45 antibody(Cat 103132), PE-Cy7 antimouse CD90.2 antibody(Cat 105326), PE antimouse CD4 antibody(Cat 11-5870-82), and APC-Cy7 antimouse CD8 antibody(Cat 100714) were used; for determining TAM1 and TAM2, PerCP-Cy5.5 antimouse CD45 antibody, PE-Cy7 antimouse CD11b antibody(Cat 101216), PE antimouse F4/80 antibody(Cat 123110), and APC antimouse CD206 antibody(Cat 141708) were used). All antibodies were diluted at a ratio of 3:100. After incubation for 30 min at room temperature, cells were washed twice with PBS for 5 min at 400× *g* at room temperature and the supernatant was eliminated. Residual cells were resuspended in MACS buffer and subjected to flow cytometry analysis. CD4 and CD8 T cells were gated from CD45 and CD90.2 positive population and separated by CD4 and CD8 staining. TAMs were gated from CD45 and CD11b positive population and determined by F4/80. Among F4/80 positive TAMs, the CD206 positive population was determined as TAM2 and negative one as TAM1.

### 2.6. Immunohistochemistry

CT26 tumors were resected, fixed with 15% formalin overnight, and embedded in paraffin. Sections were stained with antimurine CD31 antibody (diluted to 1:100) in accordance with Cell Signaling Technology’s protocol (#77699; Cell Signaling Technology). Immunohistochemical analysis was performed using the BOND-RX automated staining platform (Leica Biosystems, Tokyo, Japan) in accordance with the manufacturer’s instructions.

### 2.7. Extraction and Quantification of FTD Incorporated in Tumor DNA

FTD/TPI and DC101 were administered into the BALB/cAJcl mice bearing CT26 on days 1 to 5 and days 1, 3, 5, and 7, respectively. On day 8, the mice were euthanized and the tumor diameters corresponding to each mouse were measured. The tumors were stored in liquid nitrogen. The genomic DNA of the CT26 tumor cells was extracted and purified using a NucleoSpin^®®^ Tissue (MACHEREY-NAGEL, Düren, Germany) in accordance with the manufacturer’s instructions. The purified DNA was completely digested with DNase I and alkaline phosphatase to the deoxyribonucleoside level (including trifluridine), as previously described [14]. The samples were analyzed via LCMS/MS analysis as follows. An aliquot comprising water (10 µL), 1 M hydrochloric acid (50 µL), and internal standard (20 µL) was added to a 100-µL sample aliquot. The mixture was extracted with 1 mL of methyl *t*-butyl ether, followed by centrifugation (15,000× *g*, 5 °C, 5 min). The supernatant was dried with nitrogen at 40 °C, and the residue was reconstituted with 0.1 mL of the mobile phase comprising 0.1% acetic acid/acetonitrile (75/25, *v*/*v*). A 5-µL aliquot of the reconstituted sample was injected into an API 4000 LC/MS/MS system (AB Sciex, Foster City, CA, USA).

### 2.8. Statistical Analysis

Statistically significant differences were identified using Student’s *t*-test to determine the amount of FTD incorporated into tumor DNA and the Aspin–Welch two-tailed *t*-test to assess antitumor activity. Statistical analyses were performed using JMP Pro 9 (SAS Institute, Cary, NC, USA) and EXSUS ver. 8.1 (CAC Exicare Corp., Osaka, Japan), respectively.

## 3. Results

### 3.1. The Murine Variant of Ramucirumab, DC101, Increases the AntiTumor Efficacy of FTD/TPI

FTD/TPI and DC101, the murine variant of ramucirumab, were administrated to BALB/cAJcl mice harboring the CT26 mouse colorectal tumor. The tumor volume (TV) and BWC in CT26 are indicated in Figure 2. On day 15, FTD/TPI and DC101 monotherapy significantly inhibited tumor growth and the tumor growth inhibition (TGI) was 38.0% and 30.6%, respectively (*p* < 0.001). Furthermore, the antitumor activity of the combination treatment group was significantly superior to that of either monotherapy groups (*p* < 0.001) and tolerability was observed in all groups from the viewpoint of body weight change that were less than 10 % loss (Table 1). The TV value of the FTD/TPI and DC101 combination therapy group was significantly lower than that of either of the monotherapy groups (*p* < 0.001).

### 3.2. Increased FTD Levels in Tumor DNA after Combination Treatment with DC101 and FTD/TPI Treatment

Combination treatment with FTD/TPI and DC101 exerted an additive effect; hence, to investigate the underlying mechanism, we performed LC/MS/MS analysis to determine the FTD levels in CT26 tumor DNA on day 8 (Figure 3). FTD levels in tumor DNA reportedly reflect its antitumor efficiency [5]. Indeed, FTD levels in CT26 tumor DNA were significantly higher upon combination treatment with FTD/TPI and DC101 than upon FTD/TPI monotherapy (*p* < 0.05, Figure 3). Although sampling on day 8 was performed 3 days after the final FTD/TPI dose on day 5, FTD levels remained higher upon combination treatment with FTD/TPI and DC101 than upon FTD/TPI monotherapy. These results indicate that combination treatment with FTD/TPI and DC101 induced FTD incorporation into tumor DNA, and the antitumor efficacy of combination therapy was associated with increased FTD levels in tumor DNA.

### 3.3. Tumoral CD31 Levels after Treatment with Antimouse VEGFR-2 Monoclonal Antibody (mAb)

To confirm the activity of DC101 against tumoral microvessels in the CT26 tumor, we evaluated CD31 expression through immunohistochemistry after DC101 monotherapy or combination treatment with FTD/TPI and DC101. On day 8, CD31 was downregulated upon both monotherapy and combination therapy compared to the control (Figure 4).

### 3.4. Abundance of TAM and Tumor-Infiltrating Lymphocytes (TILs) upon Combination Treatment with FTD/TPI and DC101

We performed flow cytometry to determine whether combination therapy with FTD/TPI and DC101 influenced the TAM and T cell population in CT26 tumors in mice. On day 8, which is the predicted time before the onset of the tumor immune response based on complete antitumor activity, using single cell suspensions of TIL prepared from the tumor tissues of the animals, multicolor staining and flow cytometry analyses were conducted. The protumor M2 type tumor-associated macrophages (TAM), called TAM2 [15], percentage of the total TAM (antitumor M1 type macrophages TAM1 and TAM2) in the FTD/TPI-treated group was lower than that of the untreated control group. However, the TAM2 population was higher upon combination treatment with FTD/TPI and DC101 than upon FTD/TPI monotherapy (Figure 5a,b, Appendix A). In addition, the CD4 and CD8 T cell percentage of the total lymphocytes were not changed in all groups except for FTD/TPI monotherapy group (Figure 5c, Appendix A). Therefore, combined antitumor activity of FTD/TPI and anti-VEGFR-2 mAb therapy did not result from its tumor immunity, at least in TAM-associated immunosuppression.

## 4. Discussion

In this study, we demonstrated for the first time that the combination of anti-VEGFR-2 mAb with FTD/TPI resulted in increased overall antitumor efficacy with tolerability, and therefore, with no changes in the treatment safety profile compared to each respective monotherapy in a mouse colorectal tumor syngeneic model.

In a previous study, the incorporation of FTD into the DNA of human CRC in the SW48 and HCT 116 models was higher in the FTD/TPI and bevacizumab combination group than that in the FTD/TPI monotherapy group [9]. Blood vessels in tumors are structurally and functionally abnormal [8]. Therefore, by normalizing the vascularization during tumor progression, the accumulation of FTD into tumor DNA can potentially be boosted. Bevacizumab is an anti-VEGF-A monoclonal antibody that targets VEGFR ligand [8], and nintedanib is a multikinase inhibitor that targets VEGFRs, platelet-derived growth factor receptors, and fibroblast growth factor receptors [16]. Ramucirumab, examined herein, is a monoclonal anti-VEGFR-2 antibody. Although these drugs have somewhat different targets, increased FTD incorporation into DNA was commonly observed after combination treatment with FTD/TPI and DC101 compared to FTD/TPI alone. Therefore, this study reinforces the hypothesis that vascular normalization increases FTD uptake. In this study, tumors were harvested 24 h after the final dose (day 7) of DC101 administered on alternate days. Furthermore, vascular endothelial cell marker CD31 was downregulated after bevacizumab treatment between days 3 and 5 and inverted on day 8 [17]. Therefore, our sampling time is reasonable, considering the inhibitory effects of tumor blood vessels. In our previous study, we collected tumors 2 h or 24 h after the final administration of FTD/TPI and 48 h after the final dose of the anti-VEGF agent [9]. In addition, CD31 was downregulated especially in FTD/TPI and DC101 combination group. Possibly, there is some unknown mechanism of interaction with the CD31 expression on its combination. However, further study is warranted to understand the reason why CD31 was substantially downregulated by the FTD/TPI and DC101 combination therapy.

However, herein, tumors were harvested 72 h after the final dose of FTD/TPI, and the increased levels of FTD in tumor DNA were sustained (Figure 3). These results indicate that FTD persists in tumors and sustained antitumor effects were observed upon combination therapy with the anti-VEGF agent. Previous studies have reported a longer FTD persistence in tumors than in highly proliferating bone marrow cells [18]. Peripheral blood mononuclear cells are considered to be derived mostly from the bone marrow. The differential turnover rates of FTD-positive tumor cells and bone-marrow-derived cells may explain the persistent tumor-specific cytotoxicity with tolerable hematological toxicity.

We used the murine CRC cell line CT26, which harbors a homozygous *KRAS* mutation at G12D and shares with aggressive, undifferentiated, and refractory human CRC cells [19], and thus contains highly severe tumor model features for the evaluation of the effectiveness of FTD/TPI. The effect of ramucirumab is not influenced by the *KRAS* tumor status among CRC patients [20]. Therefore, combination treatment with FTD/TPI and ramucirumab may be beneficial in the clinical setting regardless of the *KRAS* status. The effect of bevacizumab is not influenced by the *KRAS* status [21] and combination therapy with bevacizumab and FTD/TPI is not influenced by the *KRAS* status in the clinical setting [10]. In addition, unlike other molecular-targeted drugs, antiangiogenic agents are not required to use biomarker selection in clinical settings. Therefore, considering these indications, it seems that this study result could be generalized.

Ramucirumab reportedly activates tumor immunity and its antiangiogenic activity [22], and FTD/TPI reportedly eliminates TAM2, resulting in cytotoxic CD8+ T-cell infiltration and activation [23]. We therefore focused on the tumor immune system upon combination therapy with the anti-VEGFR2 mAb and FTD/TPI, especially in TAMs. Herein, we analyzed TAMs as a tumor immunosuppressive cell population. Generally, TAMs are divided into two phenotypes [24]: TAM1 and TAM2. TAM1 in tumor tissues stimulates tumor immunity and suppresses tumor progression. In contrast, TAM2 enhances tumor cell invasion and motility and stimulates angiogenesis, suppresses the immune response, and prevents tumor cell infiltration by natural killer and T cells [25]. In this study, a decline in TAM2 was observed only upon FTD/TPI monotherapy (Figure 2). In the human gastric cancer MKN45 xenograft NOD/SCID mouse model, DC101 monotherapy exerted antitumor effects, and combination treatment with nanoparticle albumin-bound paclitaxel (nab-PTX) improved DC101 efficacy [26]. Therefore, some tumor-immunity-independent mechanisms may have been activated upon combinational antitumor therapy with DC101 and a chemotherapy agent. However, further studies are required to assess tumor immunity.

This study shows that combination therapy with FTD/TPI and anti-VEGFR-2 antibody exerted additive effects. However, these results are based on only one cell line in the syngeneic model; therefore, the present findings warrant further validation in future studies.

In conclusion, this study is the first to report that combination therapy with FTD and the anti-VEGFR-2 antibody might be effective for colorectal and, potentially, gastric cancer. Although further investigation is required to elucidate the mechanism underlying FTD and anti-VEGFR2 antibody treatment efficacy for tumors in detail, this study shows that FTD-based treatment potentially improves clinical outcomes. A phase II study evaluating the efficacy of combination treatment of FTD/TPI and ramucirumab among patients with unresectable advanced or recurrent gastric cancer is currently ongoing in Japan (JapicCTI-194596). Therefore, these results provide evidence regarding the antitumor activity of combination therapy with FTD/TPI and ramucirumab for gastric cancer patients, and the results of this trial are much anticipated.

## 5. Conclusions

In CT26 syngeneic models, tumor volume was significantly lower upon combination treatment than upon FTD/TPI or ramucirumab murine surrogate antibody DC101 monotherapy, indicating the additive effects of FTD/TPI and DC101. DNA-incorporated FTD levels were significantly higher on FTD/TPI and DC101 combination therapy than upon FTD/TPI monotherapy. Furthermore, vascular endothelial cell-specific marker CD31 was downregulated in DC101-treated tumors. These results indicate that combination therapy with FTD/TPI and ramucirumab is a promising treatment alternative.

## Figures and Tables

**Figure 1 jcm-09-04050-f001:**
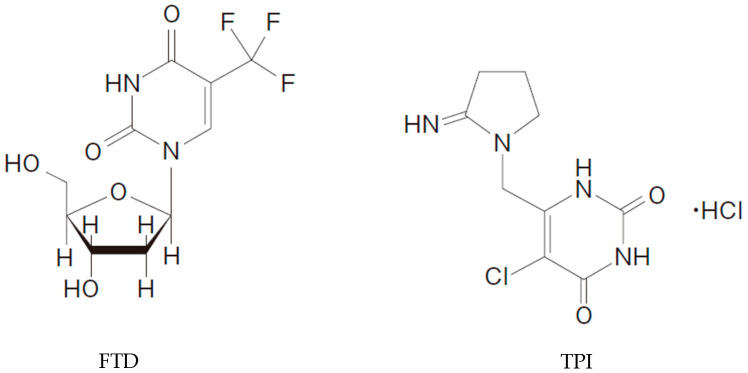
Structures of trifluridine (FTD) and tipiracil (TPI).

**Figure 2 jcm-09-04050-f002:**
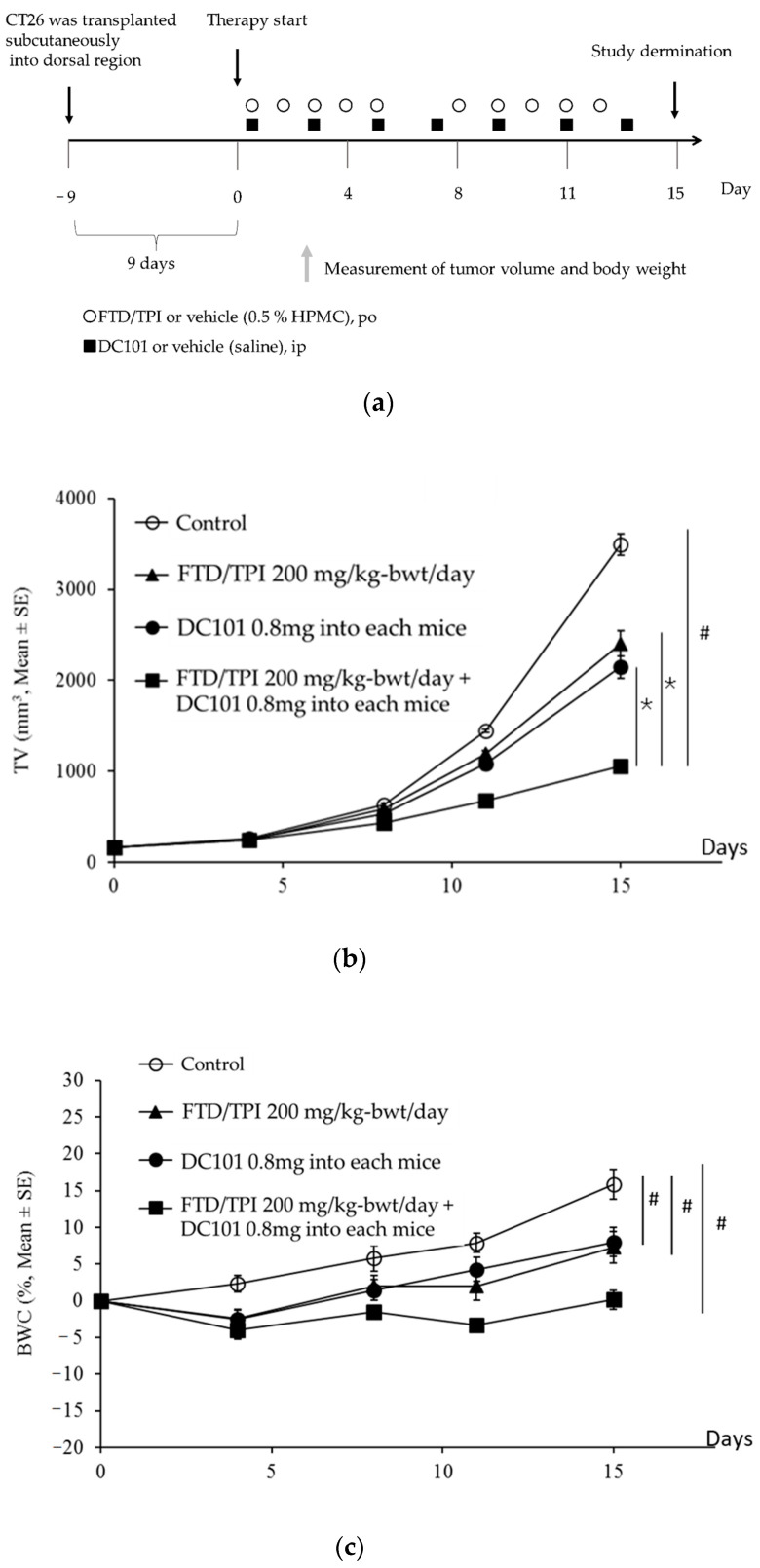
Antitumor activity of antimouse vascular endothelial growth factor receptor (VEGFR)-2 monoclonal antibody (mAb) (DC101) and FTD/TPI combination therapy in a mouse with colorectal cancer CT26 tumors. Schematic diagram of treatment (**a**), change in tumor volume (**b**) and body weight (**c**) in CT26 tumor-bearing mice. DC101 was administered intraperitoneally (ip) on days 1, 3, 5, 7, 9, 11, and 13. FTD/TPI was administered orally from days 1 to 5 and 8 to 12. Mice were treated with vehicle (◯), DC101 (0.8 mg into each mice, ip) (●), FTD/TPI (200 mg/kg-bwt/day) (▲), and DC101 (0.8 mg into each mice, ip) plus FTD/TPI (200 mg/kg-bwt/day) (■). Data are presented as means ± SE values (*n* = 6). Tumor volume and body weight were measured twice weekly. * *p* < 0.05 compared to the FTD/TPI or DC101 monotherapy group; Aspin–Welch *t*-test. ^#^
*p* < 0.05 compared to the control group; Aspin–Welch *t*-test.

**Figure 3 jcm-09-04050-f003:**
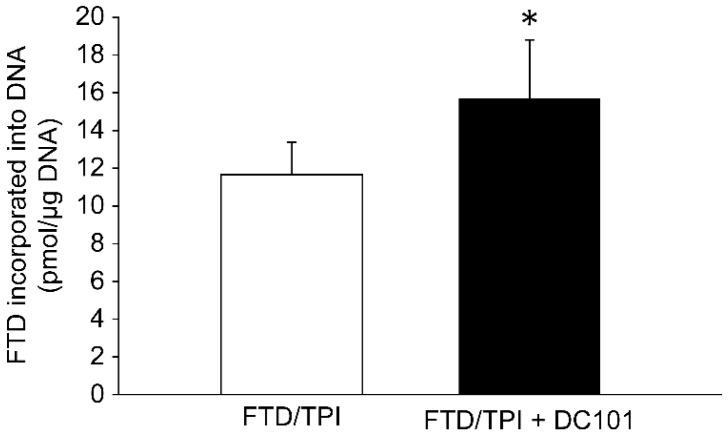
The concentration of FTD incorporated in the DNA of the mouse colorectal cancer CT26 syngeneic model. Mice were administered FTD/TPI monotherapy (200 mg/kg, orally once daily from day 1 to 5, open column) or combination therapy with FTD/TPI (200 mg/kg, orally once daily from day 1 to 5) and DC101 (0.8 mg into each mouse; ip on days 1, 3, 5, and 7; closed column). CT26 tumors were collected at 72 h following the final administration of the FTD/TPI (day 5). Data are presented as mean ± SE values (*n* = 3). * *p* <0.05 compared to the FTD/TPI monotherapy group; Student’s *t*-test.

**Figure 4 jcm-09-04050-f004:**
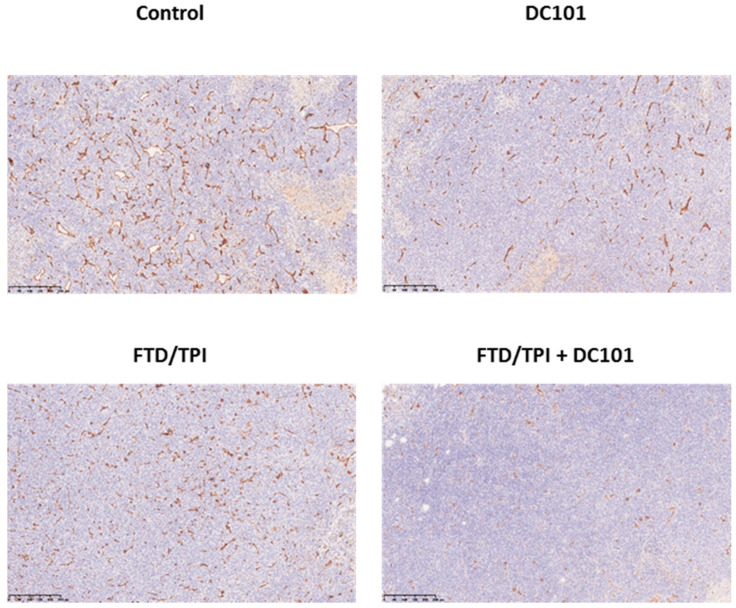
Vascular endothelial cells were visualized via immunohistochemistry using anti-CD31 antibody. FTD/TPI was administered orally from day 1 to 5 (200 mg/kg-bwt/day). DC101 was injected at 0.8 mg into each mouse; ip, on days 1, 3, 5, and 7. Combination therapy with FTD (200 mg/kg-bwt/day) and DC101 (0.8 mg into each mouse, ip) was administered in accordance with the same schedule. CT26 tumors were harvested and embedded in paraffin 24 h, followed by DC101 administration (day 7).

**Figure 5 jcm-09-04050-f005:**
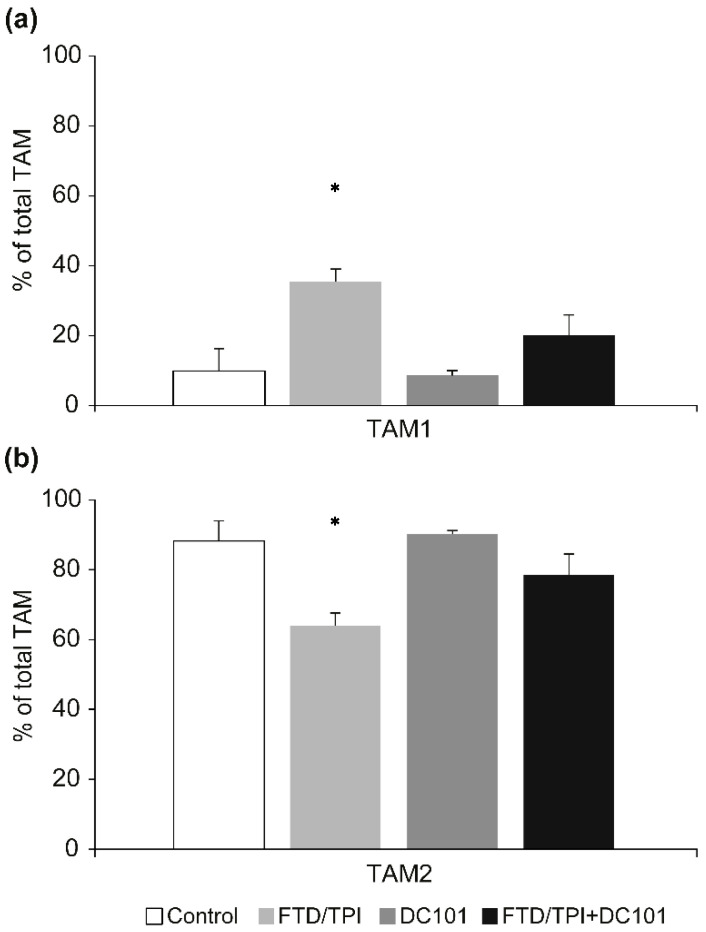
The proportion of each tumor-infiltrating lymphocytes (TIL)-infiltrated tumor-associated macrophage (TAM) subpopulation TAM1 (**a**), TAM2 (**b**), and CD4 and CD8 subpopulation (**c**) in CT26 cells was determined through flow cytometry. FTD/TPI was administered orally from day 1 to 5 (200 mg/kg-bwt/day). DC101 was administered at 0.8 mg into each mouse, ip on days 1, 3, 5, and 7 (ip). Combination therapy with FTD/TPI (200 mg/kg-bwt/day) and DC101 (0.8 mg into each mouse, ip) was administered. CT26 tumors were harvested 72 h after final drug administration (day 5). Data are presented as mean ± SE values (*n* = 3). * *p* < 0.05 compared to the control group; Student’s *t*-test.

**Table 1 jcm-09-04050-t001:** Antitumor activity and body weight changes in mice implanted with murine colorectal tumor CT26 after treatment with FTD/TPI and DC101.

Drug	Schedule Route	Number of Animals	TV ^(a)^ (mm^3^, Mean ± SE)	*p*-Value Aspin–Welch			TGI ^(b)^	BWC ^(c)^ (%, Mean ± SE)
				Control	FTD/TPI	DC101		
Control	Days 1–5 and 8–12 (sid, po) + Days 1, 3, 5, 7, 9, 11, and 13(sid, ip)	6	3491.8 ± 120.7					15.8 ± 2.0
FTD/TPI 200 mg/kg-bwt/day	Days 1–5 and 8–12 (sid, po)	6	2404.9 ± 141.8	<0.001			30.6	7.3 ± 2.2
DC101 0.8 mg into each mouse	Days 1, 3, 5, 7, 9, 11, and 13 (sid, ip)	6	2145.2 ± 123.8	<0.001			38.0	8.0 ± 2.0
FTD/TPI 200 mg/kg-bwt/day + DC101 0.8 mg into each mouse	Days 1–5 and 8–12 (sid, po) + Days 1, 3, 5, 7, 9, 11, and 13 (sid, ip)	6	1052.3 ± 67.5	<0.001	<0.001	<0.001	69.3	0.1 ± 1.3

^(a)^: Tumor volume (TV) on Day 15 was calculated according to the following formula: TV = (length) × (width)2/2, ^(b)^: Tumor growth inhibition rate (TGI) on Day 15 on the basis of relative tumor volume (RTV: tumor volume on Day 15/ tumor volume on Day 0) was calculated according to the following formula: TGI (%) = [ (mean RTV of the control group)−(mean RTV of the treated group)]/(mean RTV of the control group) × 100. ^(c)^: Body weight change (BWC, %; mean ± SE) on Day 15 were calculated according to the following formula: BWC (%) = [(BW on Day 15)−(BW on Day 0)]/(BW on Day 0) × 100. SE, standard error; *p*-value by Aspin–Welch *t* test as compared to the control and monotherapy group; TV, tumor volume.

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
