# Peer review of "Efficacy of Combination Chemotherapy Using a Novel Oral Chemotherapeutic Agent, FTD/TPI, with Ramucirumab Murine Version DC101 in a Mouse Syngeneic Cancer Transplantation Model"

_jcm, 2020, doi:10.3390/jcm9124050_

Round 1
Reviewer 1 Report
In this paper, authors presented the efficacy of a combination therapy having oral chemotherapeutic agent, FTD/TPI together with DC101 in a mouse cancer transplantation model. The study is well performed overall and presented nicely. I have few revisions:
- Please consider changing the title of the paper. Current title appears to be long and not compelling.
- In the introduction, please add a structure of Trifluridine/tipiracil and elaborate more about DC101.
- In section 2.5, Please elaborate the flow cytometry analysis methods and parameters.
- Please also elaborate Immunohistochemistry study protocol.
- In Figure 1, please add images of the animals showing tumor volume for comparison.
- Please add a conclusion section
Author Response
- Please consider changing the title of the paper. Current title appears to be long and not compelling.
Response to the comments:
We thank the reviewer for this comment and agree that title should be changed as follows.
Page 1, line2–5 (new manuscript, changes highlighted)
Efficacy of combination chemotherapeutic agent, FTD/TPI, with ramucirumab murine version DC101 in a mouse syngeneic cancer transplantation model
- In the introduction, please add a structure of Trifluridine/tipiracil and elaborate more about DC101.
Response to the comments:
We appreciate this feedback. Accordingly, our manuscript has been revised as follows.
Page 1, line34–35 (new manuscript, changes highlighted)
The structure of FTD/TPI was depicted in Figure. 1.
Page 2, line67–68 (new manuscript, changes highlighted)
and used as a surrogate antibody for ramucirumab according to the Eli Lilly’s interview form,
Figure 1. was added in page 5.
- In section 2.5, Please elaborate the flow cytometry analysis methods and parameters.
Response to the comments:
Thank you for providing these comments. Accordingly, we have revised the flow cytometry analysis methods as follows.
Page 3, line119–125 (new manuscript, changes highlighted)
antibodies (biolegend) were added (for determining CD4, CD8 T cells, PerCP-Cy5.5 anti-mouse CD45 antibody(Cat 103132), PE-Cy7 anti-mouse CD90.2 antibody(Cat 105326), PE anti-mouse CD4 antibody(Cat 11-5870-82), and APC-Cy7 anti-mouse CD8 antibody(Cat 100714) were used, and for determining TAM1 and TAM2, PerCP-Cy5.5 anti-mouse CD45 antibody, PE-Cy7 anti-mouse CD11b antibody(Cat 101216), PE anti-mouse F4/80 antibody(Cat 123110), and APC anti-mouse CD206 antibody(Cat 141708) were used). All antibodies were diluted at a ratio of 3:100.
Page 3, line127–130 (new manuscript, changes highlighted)
CD4, CD8 T cells were gated from CD45 and CD90.2 positive population and separated by CD4 and CD8 staining. TAM were gated from CD45 and CD11b positive population and determined by F4/80. Among F4/80 positive TAM, CD206 positive population was determined as TAM2 and negative one as TAM1.
- Please also elaborate Immunohistochemistry study protocol.
Response to the comments:
We thank the reviewer for this suggestion. According to this comment, we revised Immunohistochemistry study protocol as follows.
Page 3, line133–135 (new manuscript, changes highlighted)
fixed with 15% formalin overnight, and embedded in paraffin. Sections were stained with anti-murine CD31 antibody (diluted to 1:100) in accordance with Cell Signaling Technology’s protocol (#77699; Cell Signaling Technology).
- In Figure 1, please add images of the animals showing tumor volume for comparison.
Response to the comments:
We thank the reviewer for this suggestion. Unfortunately, we don’t have tumor images in this study.
- Please add a conclusion section
Response to the comments:
We thank the reviewer for this comment and according to this comment, we have added a conclusion part as follows.
Page 10, line249–256 (new manuscript, changes highlighted)
- Conclusion
In CT26 syngeneic models, tumor volume was significantly lower upon combination treatment than upon FTD/TPI or ramucirumab murine surrogate antibody DC101 monotherapy, indicating the additive effects of FTD/TPI and DC101. DNA-incorporated FTD levels were significantly higher on FTD/TPI treatment and combination therapy than upon FTD/TPI monotherapy. Furthermore, vascular endothelial cell-specific marker CD31 was downregulated in DC101-treated tumors. These results indicate that combination therapy with FTD/TPI and ramucirumab is a promising treatment alternative.

Reviewer 2 Report
- Antibodies dilutions and cat. numbers should be added to the manuscript.
- This paper is quite well written
Author Response
- Antibodies dilutions and cat. numbers should be added to the manuscript.
Response to the comments:
We would like to express our deepest appreciation to the reviewer for this comment; we have revised the manuscript based on your comments as follows.
Page 3, line119–125 (new manuscript, changes highlighted)
antibodies (biolegend) were added (for determining CD4, CD8 T cells, PerCP-Cy5.5 anti-mouse CD45 antibody(Cat 103132), PE-Cy7 anti-mouse CD90.2 antibody(Cat 105326), PE anti-mouse CD4 antibody(Cat 11-5870-82), and APC-Cy7 anti-mouse CD8 antibody(Cat 100714) were used, and for determining TAM1 and TAM2, PerCP-Cy5.5 anti-mouse CD45 antibody, PE-Cy7 anti-mouse CD11b antibody(Cat 101216), PE anti-mouse F4/80 antibody(Cat 123110), and APC anti-mouse CD206 antibody(Cat 141708) were used). All antibodies were diluted at a ratio of 3:100.
- This paper is quite well written
We thank the reviewer for this comment sincerely.

Reviewer 3 Report
Resistance to chemotherapy is one of the major failures of current chemotherapy treatment. The current study tried to demonstrate the therapeutic efficacy of combined treatment of FTD/TPI with DC101 in murine CT26 colon cancer. This manuscript has scientific merits, however, it has scope for revision at present and can be considered after revision.
Line 20 -23, abstract: Need to rephrase this sentence, "Tumor growth inhibition......combination therapy".
Line 91, 2.3. Please provide the location of the tumor inoculated into mice?
Line 96, 2.4. What was the size of the tumor, when the treatment was started? It is not clear in the method section that whether the treatment was given at the time of tumor inoculation or, the tumor was allowed to reach at certain size and further animals have received treatment? It would be better to add a schematic figure for the treatment regime in figure 1.
Line 96. Please provide a brief description of the effective dose selection of FTD/TPI/DC101 from reference.
Line 114, 2.5. Please provide a positive control for CD45 stain in the figure.
Line 119, 2.5. Please provide details of the data calculation of determining the % total TAM /total leukocytes count. How did the author calculate the TAM population isolated from the tumor? is it the total number of events counted through flow cytometry? Please provide the population selection graph plot in the figure or in the supplement.
Line 148, 3. Results. Did the author determine EC50 of FTD/TPI and DC101 in the colon cancer cell lines CT26 invitro? Please provide relevant information on the cytotoxic effect of compound(s) in cancer cells?
Line 169, 3.2. The author found significant incorporation of FDT into cancer cell DNA. It would be interesting to see the effect on DNA damage. I would suggest determining gamma H2AX expression in these experimental tumors.
Line 173, 3.3. Please provide IgG isotype control figure 3.
Line 193, Figure: 1a . there is a need to add and compare the effect relative to mice received vehicle control 0.5% hydroxypropyl methylcellulose (HPMC) solution.
Chemo-toxicity is one of the major side effects of chemotherapy treatment. Figure 1b. BWC data suggesting some kind of toxicity in mice upon receiving FTD/TPI+DC101. Did the author observes and determine any drug toxicity study in these mice? Please recalculate and provide the p-value in BWC relative to Control at day 15. Please provide data in S.E.M. for Figures 1a & B.
CD4/CD8 labeling/staining protocol is missing in the material method section. Please provide detailed isolation and staining procedure and also provide isotype used for the characterization of immune cells.
Line 221, Figure 4. Please provide the CD4+/CD8+ population in the quadrant plot. Please annotate and describe the the figure in the footnotes properly (a,b,c?). Please indicate the p-value difference relative to control (vehicle control) and FTD/TPI.
There is an extensive need for statistical analysis throughout the manuscript. The author needs to be consistent with one unit form in figures/results and uses the same throughout the manuscript. Present data in S.E.M.
Author Response
Response to the comments:
We thank the reviewer for this comment and agree that this should be modify the abstract; we have rephrase as follows.
Page 1, line20–23 (old manuscript)
Tumor growth inhibition (TGI) on day 15 was 38.0% and 30.6% upon DC101 monotherapy (0.8 mg, ip, days 1, 3, 5, 7, 9, 11, and 13) and FTD/TPI monotherapy (200 mg/kg/d, po, days 1–5 and days 8–12), respectively, and 60.3% upon combination therapy.
Page 1, line20–22 (new manuscript, changes highlighted)
Tumor growth inhibition (TGI) on day 15 was 38.0% and 30.6% upon DC101 monotherapy and FTD/TPI monotherapy respectively, and 60.3% upon combination therapy.
Line 91, 2.3. Please provide the location of the tumor inoculated into mice?
Response to the comments:
We thank the reviewer for this comment. Accordingly, we have added the location information as follows.
Page 2, line 92 (new manuscript, changes highlighted)
transplanted subcutaneously into dorsal region of each mouse.
Line 96, 2.4. What was the size of the tumor, when the treatment was started? It is not clear in the method section that whether the treatment was given at the time of tumor inoculation or, the tumor was allowed to reach at certain size and further animals have received treatment? It would be better to add a schematic figure for the treatment regime in figure 1.
Response to the comments:
We appreciate this feedback. Accordingly, our manuscript has been added information as follows and modified figure 2.
Page 3, line 98-99 (new manuscript, changes highlighted)
Treatment groups contained a similar range of initial tumor volumes (50 to 200 mm3at day 0).
Line 96. Please provide a brief description of the effective dose selection of FTD/TPI/DC101 from reference.
Response to the comments:
We apologize for not carefully checking our explanation for effective dose selection and references previously. We thus revised the references as follows.
Page 3, line 97-98 (new manuscript, changes highlighted)
at the reported effective dose (200 mg/kg/d) [12]. DC101 was administered intraperitoneally (ip) at the reported effective dose of 0.8 mg on days 1, 3, 5, 7, 9, 11, and 13 [13].
[Ref12] Kataoka Y, Iimori M, Fujisawa R, Morikawa-Ichinose T, Niimi S, Wakasa T, Saeki H, Oki E, Miura D, Tsurimoto T, Maehara Y, Kitao H. DNA Replication Stress Induced by Trifluridine Determines Tumor Cell Fate According to p53 Status. Mol Cancer Res. 2020 Sep;18(9):1354-1366.
[Ref13] Zafarnia S et al. Nilotinib Enhances Tumor Angiogenesis and Counteracts VEGFR2 Blockade in an Orthotopic Breast Cancer Xenograft Model with Desmoplastic Response. Neoplasia. 2017 19(11):896-907.olorectal and gastric cancer Oncol Lett. 2017 14(1):639-646.
Line 114, 2.5. Please provide a positive control for CD45 stain in the figure
Response to the comments:
Thank you for providing this comment. Accordingly, we have added CD45 population in supplementary figure.1. We used CD45 capturing microbeads therefore all analyzing singlet cells were almost CD45 positive (above 70%).
Line 119, 2.5. Please provide details of the data calculation of determining the % total TAM /total leukocytes count. How did the author calculate the TAM population isolated from the tumor? is it the total number of events counted through flow cytometry? Please provide the population selection graph plot in the figure or in the supplement.
Response to the comments:
We thank the reviewer for this insight and suggestion. According to this comment, we provided the population selection graph plot in the supplementary figure. 2.
Line 148, 3. Results. Did the author determine EC50 of FTD/TPI and DC101 in the colon cancer cell lines CT26 invitro? Please provide relevant information on the cytotoxic effect of compound(s) in cancer cells?
Response to the comments:
We appreciate this feedback. We did not examine CT26 in vitro cytotoxic effect of FTD and DC101. However, previous report showed in vitro FTD cytotoxicity in the CT26 up to a concentration of 100 mMa).
[Ref a] Limagne E, Thibaudin M, Nuttin L, Spill A, Derangère V, Fumet JD, Amellal N, Peranzoni E, Cattan V, Ghiringhelli F. Trifluridine/Tipiracil plus Oxaliplatin Improves PD-1 Blockade in Colorectal Cancer by Inducing Immunogenic Cell Death and Depleting Macrophages. Cancer Immunol Res. 2019 Dec;7(12):1958-1969.
Line 169, 3.2. The author found significant incorporation of FDT into cancer cell DNA. It would be interesting to see the effect on DNA damage. I would suggest determining gamma H2AX expression in these experimental tumors.
Response to the comments:
We would like to express our appreciation to the reviewer for this comment; we did not have gamma H2AX expression data so we would like to examine that in the next study. We are grateful for your suggestion.
Line 173, 3.3. Please provide IgG isotype control figure 3.
Response to the comments:
We thank the reviewer for this comment. Unfortunately we don't have IgG isotype control but we used the Cell Signaling Technology -verified protocol for which optimal staining conditions were examined by Leica BOND RX.
Line 193, Figure: 1a . there is a need to add and compare the effect relative to mice received vehicle control 0.5% hydroxypropyl methylcellulose (HPMC) solution.
Response to the comments:
We thank the reviewer for this insightful comment and suggestion. We examined vehicle control in only one group that 0.5% hydroxypropyl methylcellulose (HPMC) solution administered orally plus 0.1 mL of saline injected intraperitoneally in this study. We would like to set respective vehicle control group in the future study.
Chemo-toxicity is one of the major side effects of chemotherapy treatment. Figure 1b. BWC data suggesting some kind of toxicity in mice upon receiving FTD/TPI+DC101. Did the author observes and determine any drug toxicity study in these mice? Please recalculate and provide the p-value in BWC relative to Control at day 15. Please provide data in S.E.M. for Figures 1a & B.
Response to the comments:
Thank you for your suggestion. According to this comment, in supplementary table1, we provide the p-value in BWC relative to Control at day 15 and revised manuscript’s S.D to S.E.M. for Figures 1a & b.
CD4/CD8 labeling/staining protocol is missing in the material method section. Please provide detailed isolation and staining procedure and also provide isotype used for the characterization of immune cells.
Response to the comments:
We thank the reviewer for this comment and agree that method should be added information to the Experimental Section; we have added this as follows. However, we didn’t set isotype control in this study and unstained sample were used for characterization of immune cells.
Page 3, line 119-125 (new manuscript, changes highlighted)
antibodies (biolegend) were added (for determining CD4, CD8 T cells, PerCP-Cy5.5 anti-mouse CD45 antibody(Cat 103132), PE-Cy7 anti-mouse CD90.2 antibody(Cat 105326), PE anti-mouse CD4 antibody(Cat 11-5870-82), and APC-Cy7 anti-mouse CD8 antibody(Cat 100714) were used, and for determining TAM1 and TAM2, PerCP-Cy5.5 anti-mouse CD45 antibody, PE-Cy7 anti-mouse CD11b antibody(Cat 101216), PE anti-mouse F4/80 antibody(Cat 123110), and APC anti-mouse CD206 antibody(Cat 141708) were used). All antibodies were diluted at a ratio of 3:100.
Line 221, Figure 4. Please provide the CD4+/CD8+ population in the quadrant plot. Please annotate and describe the figure in the footnotes properly (a,b,c?). Please indicate the p-value difference relative to control (vehicle control) and FTD/TPI.
Response to the comments:
We thank the reviewer for this comment and according to this comment, we have added CD4+/CD8+ population quadrant plot.in the supplementary figure. 3 and revised footnotes properly as follows. And we denote p-value relative to control.
Figure 5. The proportion of each TIL-infiltrated tumor-associated macrophage (TAM) subpopulation TAM1 (a), TAM2 (b), and CD4 and CD8 subpopulation (c) in CT26 cells was determined through flow cytometry. FTD/TPI was administered orally from day 1 to 5 (200 mg/kg/d). DC101 was administered at 0.8 mg/body on days 1, 3, 5, and 7 (ip). Combination therapy with FTD/TPI (200 mg/kg/day) and DC101 (0.8 mg/body) was administered. CT26 tumors were harvested 72 h after final drug administration (day 5). Data are presented as mean ± SE values (n = 3). *P <0.05 compared to the control group; Student’s t-test.
There is an extensive need for statistical analysis throughout the manuscript. The author needs to be consistent with one unit form in figures/results and uses the same throughout the manuscript. Present data in S.E.M.
Response to the comments:
We thank the reviewer for pointing this out. Accordingly, we have corrected statistical analysis unit for S.E.M.

Round 2
Reviewer 3 Report
Manuscript is substantially improved. Though Few minor corrections needs to done before acceptance.
Conclusion should be placed after discussion.
line 98: please confirm whether DC101 was given ip at 0.8 mg (?) on day 1,3,5,7,9,11 and 13 ? Was it 0.8mg for all mice regardless of their body weight?
DC101 0.8 mg: it is misleading, while reading DC101 0.8 mg/body, instead can be modified “DC101 0.8mg into each mice”. If it has not given as mg/kg bwt. Describe the route/dose in appropriate unit under figure footnote and method section.
Figure 2: it is still quite confusing to understand dose scheme. Please provide schematic diagram of treatment, such as tumor inoculation, treatment start day, intermittent doses, terminal day etc.
Figure 2: FTD/TPI 200 mg/kg/day is misleading and can be presented as mg/kg-bwt/day and describe in the footnotes.
Author needs to use one abbreviation style and consistent with one unit throughout the manuscript. Example line 233: mg/kg/d, either write ‘d’ or ‘day’ and should be consistent through out the manuscript and correct other similar points before the resubmission.
Author Response
Comments and Suggestions for Authors
Manuscript is substantially improved. Though Few minor corrections needs to done before acceptance.
Conclusion should be placed after discussion.
Response to the comments:
We thank for this comment and agree that conclusion was be placed after discussion. Any revisions were highlighted using the "Track Changes" function in Microsoft Word.
line 98: please confirm whether DC101 was given ip at 0.8 mg (?) on day 1,3,5,7,9,11 and 13 ? Was it 0.8mg for all mice regardless of their body weight?
DC101 0.8 mg: it is misleading, while reading DC101 0.8 mg/body, instead can be modified “DC101 0.8mg into each mice”. If it has not given as mg/kg bwt. Describe the route/dose in appropriate unit under figure footnote and method section.
Response to the comments:
Thank you for providing these comments. Accordingly, we have revised the DC101 unit “DC101 0.8mg into each mice” under figure footnote and method section as follows.
Page 3, line98 (new manuscript, changes highlighted)
dose of 0.8 mg into each mice on days 1, 3, 5, 7, 9, 11, and 13 [13].
Figure 2: it is still quite confusing to understand dose scheme. Please provide schematic diagram of treatment, such as tumor inoculation, treatment start day, intermittent doses, terminal day etc.
Response to the comments:
We appreciate this feedback. Accordingly, figure.2 has been revised and attached schematic diagram.
Figure 2: FTD/TPI 200 mg/kg/day is misleading and can be presented as mg/kg-bwt/day and describe in the footnotes.
Response to the comments:
Thank you for providing this comment. Accordingly, we revised Figure 2 and in the footnotes: FTD/TPI 200 mg/kg/day to FTD/TPI 200 mg/kg-bwt/day.
Author needs to use one abbreviation style and consistent with one unit throughout the manuscript. Example line 233: mg/kg/d, either write ‘d’ or ‘day’ and should be consistent through out the manuscript and correct other similar points before the resubmission.
Response to the comments:
We thank for pointing this out. Accordingly, we have corrected ‘d’ to ‘day’ throughout the manuscript.
